# Stacking with Recursive Feature Elimination-Isolation Forest for classification of diabetes mellitus

**Nur Farahaina Idris** [1], **Mohd Arfian Ismail** [1,2]*, **Mohd Izham Mohd Jaya**[1], **Ashraf Osman Ibrahim**[3], **Anas W. Abulfaraj**[4], **Faisal Binzagr**[5]

**1** Faculty of Computing, Universiti Malaysia Pahang Al-Sultan Abdullah, Pekan, Pahang, Malaysia, **2** Centre of Excellence for Artificial Intelligence & Data Science, Universiti, Al-Sultan Pahang, Lebuhraya Tun Razak, Gambang, Malaysia, **3** Creative Advanced Machine Intelligence Research Centre, Faculty of Computing and Informatics, Universiti Malaysia Sabah, Sabah, Malaysia, **4** Department of Information Systems, King Abdulaziz University, Rabigh, Saudi Arabia, **5** Department of Computer Science, King Abdulaziz University, Rabigh, Saudi Arabia

* arfian@ump.edu.my

**Data Availability Statement:** The datasets used in this research project are the PIMA Indians Diabetes and Diabetes Prediction. Both datasets were acquired from the Kaggle website and are publicly

## Abstract

Diabetes Mellitus is one of the oldest diseases known to humankind, dating back to ancient Egypt. The disease is a chronic metabolic disorder that heavily burdens healthcare providers worldwide due to the steady increment of patients yearly. Worryingly, diabetes affects not only the aging population but also children. It is prevalent to control this problem, as diabetes can lead to many health complications. As evolution happens, humankind starts integrating computer technology with the healthcare system. The utilization of artificial intelligence assists healthcare to be more efficient in diagnosing diabetes patients, better healthcare delivery, and more patient eccentric. Among the advanced data mining techniques in artificial intelligence, stacking is among the most prominent methods applied in the diabetes domain. Hence, this study opts to investigate the potential of stacking ensembles. The aim of this study is to reduce the high complexity inherent in stacking, as this problem contributes to longer training time and reduces the outliers in the diabetes data to improve the classification performance. In addressing this concern, a novel machine learning method called the Stacking Recursive Feature Elimination-Isolation Forest was introduced for diabetes prediction. The application of stacking with Recursive Feature Elimination is to design an efficient model for diabetes diagnosis while using fewer features as resources. This method also incorporates the utilization of Isolation Forest as an outlier removal method. The study uses accuracy, precision, recall, F1 measure, training time, and standard deviation metrics to identify the classification performances. The proposed method acquired an accuracy of 79.077% for PIMA Indians Diabetes and 97.446% for the Diabetes Prediction dataset, outperforming many existing methods and demonstrating effectiveness in the diabetes domain.

available. Access the datasets directly from Kaggle using the following links: • PIMA Indians Diabetes dataset: https://www.kaggle.com/datasets/uciml/pima-indians-diabetes-database • Diabetes Prediction dataset: https://www.kaggle.com/datasets/iammustafatz/diabetes-prediction-dataset Please note that the datasets' availability and terms of use are governed by Kaggle's data usage policies and the original data contributors. As these datasets are publicly available, there is no need to request access from the authors.

**Funding:** : The study was supported by the Fundamental Research Grant (RDU) with vot. No. RDU220304 from Universiti Malaysia Pahang Al-Sultan Abdullah Role of Funders: The funder had no role in study design, data collection and analysis, decision to publish, or preparation of the manuscript.

**Competing interests:** The authors have declared that no competing interests exist.

## 1. Introduction

Diabetes Mellitus, generally known as diabetes, is a chronic metabolic disease and a severe epidemic that significantly rises each year, causing problems for healthcare providers around the world [1]. The history of diabetes dates back to ancient Egypt about 3000 years ago, and its impact remains profound in modern times, continuing to be a serious health concern. Hence, rigorous studies about diabetes were made. Notably, the rigid distinction between type 1 and type 2 diabetes was only established in 1936 [2]. Type 2 diabetes is the most common type of diabetes suffered by patients caused by insulin resistance [3]. Meanwhile, type 1 diabetes is caused by the destruction of β cells in the pancreas, causing the absence of insulin secretion [4]. The factors of type 1 are due to genetics. Still, they can also be caused by other external factors like low vitamin D levels, prenatal exposure to pollutants, and poor hygiene, making it easier for children to get affected [4]. Hence, this disease affected not only the aging population but also children. This disease is known as a silent illness, as many individuals who suffer from it are unaware of their condition due to unnoticeable symptoms [5]. Hence, the results of having diabetes are often too sudden, causing the individuals to have difficulty in following the treatment and pursuing lifestyle changes [6]. It is concerning because individuals with diabetes may get various health complications such as kidney disease, stroke, coronary heart disease, retinopathy [7], and emerging complications like cancer and liver disease [8]. Hence, creating awareness about the importance of rapid diagnosis in managing diabetes is crucial [9]. Early detection enables timely intervention and helps individuals with diabetes take proactive steps to improve their health and well-being. Rapid diagnosis not only facilitates better disease control but also reduces the risk of complications associated with diabetes. Each year, many studies and initiatives were implemented to assist physicians in achieving rapid and accurate diagnoses.

Back then, physicians traditionally diagnosed patients with diabetes by checking all the symptoms attentively, prolonging the diagnosis process. Nevertheless, as evolution happens, humankind has started integrating computer technology with the healthcare system. The utilization of artificial intelligence assists healthcare providers to be more efficient in the diagnosis of patients, better healthcare delivery, and more patient eccentric [10]. Therefore, it would benefit everyone regardless of socioeconomic and geographical location. The study on the integral subdivisions of artificial intelligence, which are predominantly centered around classification and data mining, is being conducted. Among the advanced data mining techniques such as deep learning and ensembles, including XGBoost and bagging, stacking emerges as one of the most prominent methods extensively employed in the diabetes domain [11, 12]. Hence, this study investigates the potential of stacking. It has been identified that stacking is a high-complexity model that leads to high computational efficiency and affects training time. The study aims to develop a lower-complexity novel stacking method for diabetes prediction that demonstrates high efficacy while employing fewer features. Thus, the study delves into utilizing stacking, the Isolation Forest as an outlier detection method, and the Recursive Feature Elimination (RFE) as the feature selection method to solve this problem. Hence, a novel method, Stacking with RFE and Isolation Forest (SRFEI), is being proposed. The contribution of this study is divided into several components:

- The application of RFE with stacking to reduce the number of features and complexity.

- Utilization of Isolation Forest with stacking removes the outlier and reduces the stacking ensemble's high complexity.

- Propose a method using stacking, RFE, and Isolation Forest that can acquire high classification accuracy.

To ensure better readability, this paper will be organized as follows: a review of literature, method, results, discussion, and finally, a conclusion.

## 2. Review of literature

### 2.1 Classification

Classification is a data mining technique that involves a knowledge extraction process [13]. Classification can be referred to as categorizing the indeterminate data into discrete classes or categories (target variables) based on specific features and variables [14–16]. Meanwhile, it is for regression tasks involving continuous features on the target variable. Many prominent methods have been applied for diabetes classification, such as J48, random forest, and Naïve Bayes [17]. However, it may lie where the inability to obtain excellent results due to big data, outliers, and discrepancy features has become a constraint [18]. Another difficulty that might develop during classification can include instability, overfitting, and underfitting issues, which cause low accuracy performance. Support Vector Machine and Decision tree classification as single classifiers acquired 81.1% and 81.3% accuracy, consecutively with the diabetes dataset collected from the Department of Medical Services, Bangkok, between 2019 and 2021 [19]. The results demonstrated a reasonable level of performance. However, there is potential for further enhancement by implementing advanced techniques. Thus, well-known data mining techniques such as Logistic Regression, Decision Tree, and Support Vector Machine are integrated with the ensemble to enhance classification performance. Notably, ensemble methods combine the predictions of multiple individual models to create a more accurate prediction.

### 2.2 Ensemble

The ensemble methods computational learning approaches are analogous to human behavior, seeking several perspectives before making any critical decisions [20]. Just like humans take diverse viewpoints to get a better conclusion, ensemble methods combine the predictions of multiple models to create a more accurate prediction. Normally, ensemble methods consist of techniques such as the Random Forest, stacking, bagging, and boosting, which work as advanced methods to improve the performance of machine learning algorithms. All the ensemble methods utilize the multiple classifiers system in which the classifiers may consist of the same type of machine learning algorithms (homogenous ensemble) or different types of algorithms (heterogeneous ensemble) [21, 22]. Existing studies elucidate that the ensembles such as AdaBoost and bagging obtained superior classification accuracies of 75.32% and performed better than single classifiers like a decision tree with 71.42% and K-Nearest Neighbor with 71.92% when classifying the PIMA Indians Diabetes dataset since the errors of the classifiers are negatively associated [23]. When multiple models are combined, it will capture the patterns of the data better. Hence, it leads to better classification performance and avoids overfitting issues [15]. There are several other theories that explain the effectiveness of ensemble methods in various sectors. For example, Allwein, Schapire, and Singer provided a deep theoretical analysis highlighting that the ensemble methods can enhance the generalization ability within the framework of large margin classifiers and demonstrating that ensembles act as a form of regularization, akin to the regularization techniques employed in single classifiers like Support Vector Machine [24, 25]. Meanwhile, Breiman explained that performance improvement is related to bias and variance [26]. The present study focused on the stacking ensemble since this method is more commonly implemented in diabetes studies and performs superiorly compared to homogenous ensembles like AdaBoost [12] and Random Forest [27], showing the potential of stacking in classification. Aside from that, stacking allows diversity that can lead to increased predictive accuracy aside from handling noisy and outlier data [28].

## 2.3 Stacking ensemble

Stacking is an ensemble method that fuses multiple classification models consisting of base and meta-classifiers [29]. This method unifies various machine-learning methods and shares similarities with other ensemble approaches, such as bagging and boosting [30]. Stacking's architecture is divided into two stages: level 0 and level 1 [31]. In level 0, base classifiers are trained using the whole training set, and each base classifier conducts classification on the data and generates its predictions. In contrast, in level 1, the meta-classifier takes the outputs or predictions the base classifiers produce as its input features. To be clear, the meta-classifier learns from the predictions made by the base classifiers rather than the original input data [32, 33]. The goal of the meta-learner is to combine these predictions effectively, considering the strengths and weaknesses of each base classifier, in order to make the final ensemble classification [34]. Stacking delivers superior performance compared to boosting and bagging when it is maximumly optimized [35]. This is primarily due to its capacity to combine multiple base classifiers using a meta-classifier, capturing diverse patterns and behaviors and ultimately enhancing the ensemble's classification capabilities. The stacking ensemble managed to obtain the classification accuracy, precision, and F1-measure of 78.2%, 72.2%, and 59.4% consecutively when using the PIMA Indians Diabetes dataset, outperforming the AdaBoost and Multi-layer Perceptron [12]. Another study shows that the stacking method acquired an outstanding accuracy of 94.48% when classifying the Saudi Arabian diabetes dataset collected from King Fahad University Hospital [36]. Hence, it shows the potential of this method in this domain. Nevertheless, stacking that generates multiple base classifiers and consists of two levels would cause higher complexity for the model and simultaneously increase the duration of training time [37, 38]. Hence, to alleviate this problem, the study utilizes the well-known feature selection technique, RFE, that is commonly employed in diabetes research.

## 2.4 RFE feature selection

One of the essential keys that has a direct impact on classification performances is the feature selection technique [39]. Feature selection eliminates unnecessary data by removing the less relevant features [40]. Many feature selection types, such as ReliefF, mutual information, and embedded methods like Lasso and Ridge, work well to reduce the number of features during classification. Still, this study focuses on the RFE as it can handle numeric and categorical data and is a more model-centric approach. The model-centric approach means that the RFE evaluates the model's performance when selecting the features, as the method would assess the impact of removing each feature in each iteration [41]. The implementation of RFE starts by training a model using all of the features. Any machine learning algorithm, including Decision Tree, Support Vector Machines, and Linear Regression, can suitably be used to train the model. As the feature selection process develops, the original model serves as a reference point or baseline for comparison. The significance of each feature is established after the original model has been trained. This is frequently accomplished using feature importance in tree-based models or by examining the weights (coefficients) given to features in linear models. Alternatively, depending on the behavior of data, different metrics such as mutual information or correlation might be utilized. Then, the step of removing the least significant feature in the data identified in the previous stage is executed [42]. The lowest-ranked features or a predetermined number of features to be removed in each iteration can serve as the direction for this elimination procedure. Instead of removing multiple features at once, RFE eliminates just one feature at a time through iteration to assess the individual contribution of each feature. The model would then be trained again on the reduced feature set. This helps identify whether the absence of that specific feature positively or negatively impacts the model's performance. This

process is repeated iteratively until a stopping criterion is met. The stopping criterion could be selecting a predetermined number of features or when the model's performance improvement saturates. RFE with XGBoost obtained an accuracy of 90% when classifying the diabetic retinopathy dataset from China, which has 60 features [43]. The existing study shows that the Chi-Square test and RFE had relatively similar performance with 13 and 18 features, both with 100% test data accuracy [44]. However, RFE takes a shorter running time to train 18 features, which RFE takes 0.286 seconds, while the Chi-Square test takes 0.803 seconds. Hence, this unveils that RFE efficiently reduces irrelevant features and has fast computational processes. Eliminating irrelevant features is intended to decrease the complexity of the method, but this may inadvertently impact performance as it causes the method to be unable to capture essential underlying patterns due to excessive feature removal or loss of important features [45]. To strike a balance, this study implements both feature selection and outlier detection techniques. This dual strategy aims to not only reduce complexity but also enhance the overall performance and interpretability of the proposed method.

## 2.5 Isolation forest

Anomalies and outliers are the data points or observations that are significantly different and deviate from the norm value of the data. Outlier detection and removal methods are needed to ensure the relevancy of the critical information in the data, as these methods play a crucial role in eliminating and mitigating biases present in data points. The method developed by Liu, Ting, and Zhou in 2008, called the Isolation Forest [46], is selected as an outlier detection method in this study. Isolation Forest leverages the concept of isolating anomalies by constructing simple yet powerful isolation trees, thus revealing the subtle variations that distinguish outliers. The general overview of this method is that Isolation Forest starts by creating a collection of isolation trees, each designed to isolate a single outlier or a small group of similar outliers. These isolation trees are binary structures, simulating recursive partitioning. The implementation of the Isolation Forest starts with a random selection of features. Each iteration selects a random feature to serve as the splitting criterion. A random value within the range of the selected feature's values is then chosen as the splitting threshold. This randomness is essential to the algorithm's efficiency in isolating outliers. Then, the training data is recursively partitioned into two subsets based on the selected feature and threshold. This process continues until it achieves the stopping criterion: the predetermined tree depth is reached, or a subset contains a single data point. Each level of the tree represents a partition of the data space. Each level of the isolation tree represents a partition of the data space. The tree structure is binary, as each node in the tree has at most two child nodes. To identify anomalies, the Isolation Forest algorithm measures the path length from the root of the tree to a leaf node for each data point, as the data with outliers will have a shorter path length than the normal instance. The isolation score for each data point is calculated by taking the average path length across all isolation trees. Lastly, the average path length is normalized and transformed into an anomaly score. Higher anomaly scores indicate a higher likelihood of being an outlier. One of the characteristics of Isolation Forest is that it does not rely on a fixed threshold to determine the outliers' level. It offers flexibility by allowing users to set the contamination hyperparameter (outliers' level in the data). Based on the existing study, XGBoost-Isolation Forest obtained an accuracy of 87.2% when classifying the TLGS Diabetes data, showing that Isolation Forest managed to reduce outliers effectively [47].

## 3. Method

This study aims to reduce the high complexity inherent in stacking, minimize the training time and outliers in the diabetes data, and improve the classification performance. Hence, this

study introduces a novel method that combines stacking with RFE and Isolation Forest, known as the SRFEI method. This method applies both feature selection and outlier detection techniques. The first step of SRFEI's classification started with shuffling the data and the imputation process using median imputation for missing values. This is followed by the split of training and test data in which training data consists of 80% of the actual data while test data has the remaining balance of 20%. The 80% training data and 20% test data split ratio is commonly applied in the diabetes domain [48]. Next, the Isolation Forest is employed for outlier detection and removal in the data. Outliers can cause disproportionate impacts on machine learning and introduce bias in diabetes prediction. Aside from that, outliers cause machine learning methods to have difficulty capturing the underlying data patterns and relationships within the diabetes data. By constructing decision tree ensembles, the technique effectively identifies anomalies and outliers within a dataset by isolating observations that deviate significantly from the norm, allowing the outliers to be eliminated adeptly. In this study, a contamination value of 0.2 signifies that approximately 20% of the data points are expected to be outliers, indicating outliers prevalent within the data. This parameter choice is based on the assumption that outliers are relatively not uncommon while still conveying a balance in the trade-off between identifying genuine anomalies and avoiding the misclassification of average data points as outliers. The number of trees hyperparameter is set as 100 by the following rule of thumbs in Isolation Forest. The guidance from healthcare experts was also utilized to establish an appropriate range for each feature and estimate the percentage of outliers present within the diabetes data. As Isolation Forest was incorporated, any outliers could be well-removed. Hence, it would reduce the complexity of the model during classification, enhance interpretability within data patterns, and produce more accurate results.

For the following step three, feature selection through the RFE would be applied to training data to find the best features. The number of features selected would be only 80% of the original features. The decision tree algorithm is chosen as the method for training the model, and Gini impurity is selected as the metric for feature importance assessment. Gini impurity provides a measure of the effectiveness of each feature in contributing to the classification performance. The RFE process will be executed iteratively, eliminating one worst-ranking feature at a time during each iteration. The features contributing the least to the model's performance will be systematically removed, ensuring that only the most valuable features are retained for model training and classification. The features removed in training data would also be removed in test data, ensuring consistency in the model. The agenda behind the implementation of RFE is to address the challenge of dealing with a large number of features in diabetes data. As diabetes datasets often contain a lot of features, attempting classification without feature selection can increase complexity and lead to longer training times. By iteratively eliminating less informative features, the goal is to streamline the data systematically without negatively impacting classification performance. Applying RFE not only helps manage the complexity of the classification task but also reduces training times, making the overall process more computationally efficient.

The next stage of the method is divided into two crucial stages: level 0 and level 1. In level 0, the training data that previously underwent train-test splitting and feature selection will be used for classification purposes. The training data would undergo stratified $k$-fold cross-validation in which the $k$ value has been set as ten. To prevent bias, the second shuffling is done before cross-validation to introduce randomness and reduce any potential ordering bias. This study employed stratified $k$-cross-validation to avoid overfitting. One-fold from the cross-validation would become the validation set to assess the method's performance after executing the classification process. The remaining $k$-1 folds, basically the training sets, would undergo a classification process. After that, base classifiers would proceed to undergo the classification

process. SRFEI method applied Logistic Regression, K- Nearest Neighbor, Decision Tree, and Support Vector Machine as base classifiers. The different types of machine learning methods as the base classifiers would ensure diversity in the results, as diversity is significant in stacking [49]. All the base classifiers would make the classification on the validation set. Average weighted is used to combine predictions as it harnesses the strength of multiple models and balances the contribution of each model. The average weighted technique has the flexibility to fine-tune the weights, which can also be done based on expert assistance to enhance the performance. This strategic application aims to expedite the most accurate decision-making process. The cross-validation process continued until all the unique folds from the 10-folds had been given the opportunity to become the validation set.

Then, after the process of classification in level 0 is completed, level 1 will start. Theoretically, the idea behind level 0 is that the training data would transform relatively into new data known as secondary data, as the classes would be based on the best prediction inputs by base classifiers (acquired from validation sets prediction) and combined using the weighted average technique. In level 1, all the transformed training data from level 0 will be used for classification. This will create better underlying data patterns for classification. The meta-classifier, Logistic Regression, which follows the rule of thumb in stacking, would be used to classify the transformed data. Lastly, the classification would assist in the prediction of test data (transforming the same features as training data). The test data is independent of level 0 and any classification processes. Hence, it would provide fair assessments for the method. Fig 1 shows the architecture of SRFEI at level 0 and level 1. The application of RFE and Isolation Forest significantly reduces the potential bias, outliers, and anomalies typically in diabetes data, making the SRFEI able to understand data patterns better and simultaneously reducing the complexity of the method. This study shows the flowchart of SRFEI in Fig 2 to aid in understanding the logical flow and the pseudocode of SRFEI in Fig 3 as representations to provide clear explanations of the proposed method.

## 4. Results

The evaluation metrics employed include accuracy, precision, recall, and F1-measure, which provide comprehensive insights into the model's performance. The accuracy is the total number of correct predicted instances over the total instances [50]. Precision is the value of true positives over the sum of true and false positives. In contrast, recall is the value of true positives over the sum of true positives and false negatives. Lastly, the F1 measure is the harmonic mean of precision and recall. The measurement of training time in seconds is used as a time-related performance metric. The accuracy, precision, recall, F1 measure, and standard deviation (SD) of accuracy values are recorded in percentage format (%), while training time is in seconds (s). The results represent the average from thirty independent runs. This approach was adopted to ensure more reliable assessments of the model's performance.

### 4.1 Results of PIMA Indians diabetes and diabetes prediction datasets classification

In this study, two diabetes datasets were utilized to test the performance of the proposed method. The first dataset is the PIMA Indians Diabetes dataset. This data was originally from the National Institute of Diabetes and Digestive and Kidney Diseases. The dataset contains eight features exclusive of the target variable and 768 instances. The target variable consists of value 0 (negative test for diabetes) and value 1 (positive test for diabetes), in which 268 consists of positive tests for diabetes and 500 are negative tests for diabetes. The features in this dataset are the number of times pregnant, Glucose, Blood Pressure, Skin Thickness, Insulin, BMI,

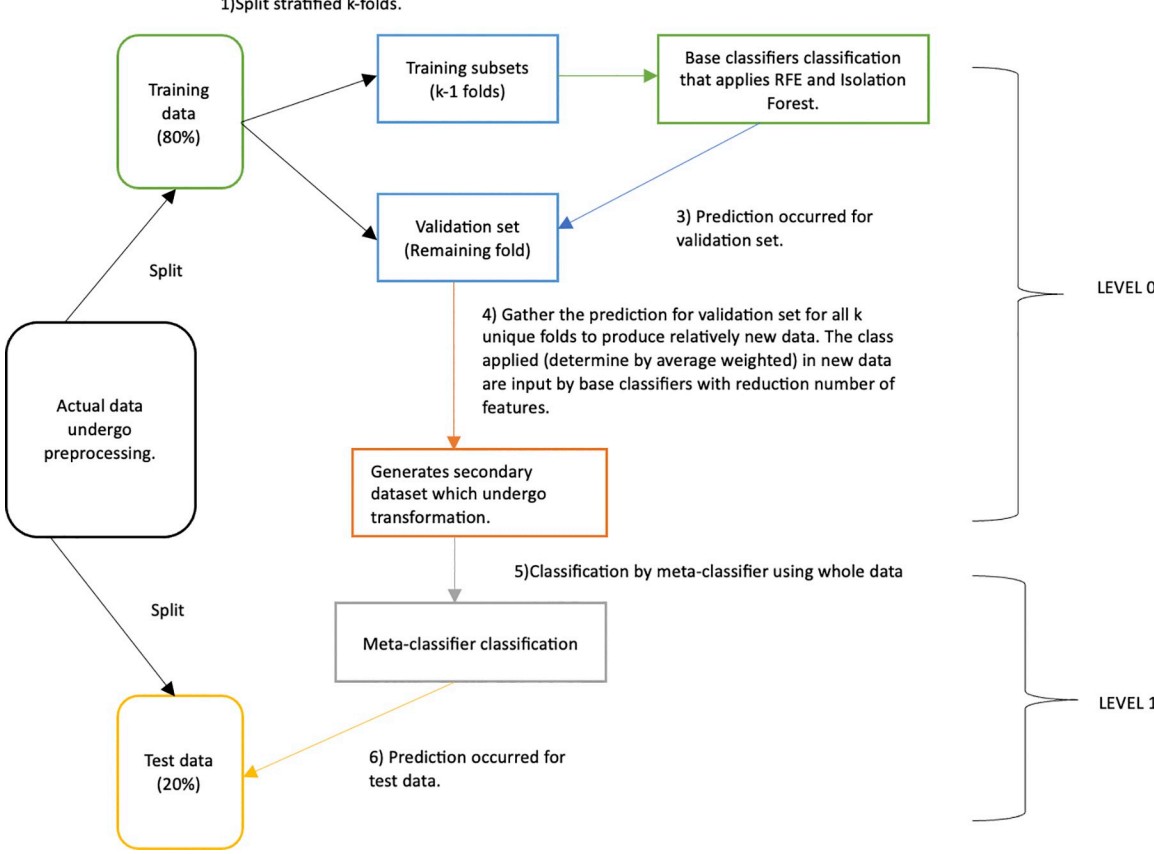

**Fig 1. The architecture of SRFEI at level 0 and level 1.**

Diabetes pedigree function, and age in years format. All those features are numeric datatypes. For this dataset, specific features such as Glucose, Blood Pressure, Skin Thickness, Insulin, BMI, and Diabetes Pedigree Function containing values of 0 were replaced with their respective median values. This preprocessing was guided by the experts' advice, considering that obtaining a score of 0 for these features is implausible and likely indicates missing data.

The second dataset is the Diabetes Prediction dataset. This dataset has 100,000 instances with eight features, not including the target variable. The target variable is binary: 0 for non-diabetes and 1 for diabetes. The distribution indicates 91,500 instances labeled with a value of 0 (non-diabetes) and the remaining 8,500 instances labeled with 1 (diabetes) for the target variable. The features in the Diabetes Prediction dataset are Gender, Age, Hypertension, Heart Disease, Smoking History, BMI, HbA1c level, and Blood Glucose. All those features are numeric datatypes except Gender and Smoking History. As both datasets are hugely imbalanced, the macro average technique would be used in the experiment in order to assess the minority class (diabetes) accurately and to ensure that the minority class is not overshadowed by the majority class. Tables 1 and 2 show the results of the PIMA Indians Diabetes and Diabetes Prediction datasets classification results using SRFEI and conventional stacking.

The SRFEI method uses six features in the experiments for both datasets, as the method selected only 80% of the datasets' original features. Based on the experimentation, a significant reduction in training time for the PIMA Indians Diabetes dataset classification was observed, dropping from 4.286 seconds with conventional stacking to just 2.646 seconds when using the RFE. Moreover, the application of the RFE was also beneficial in the prediction aspect,

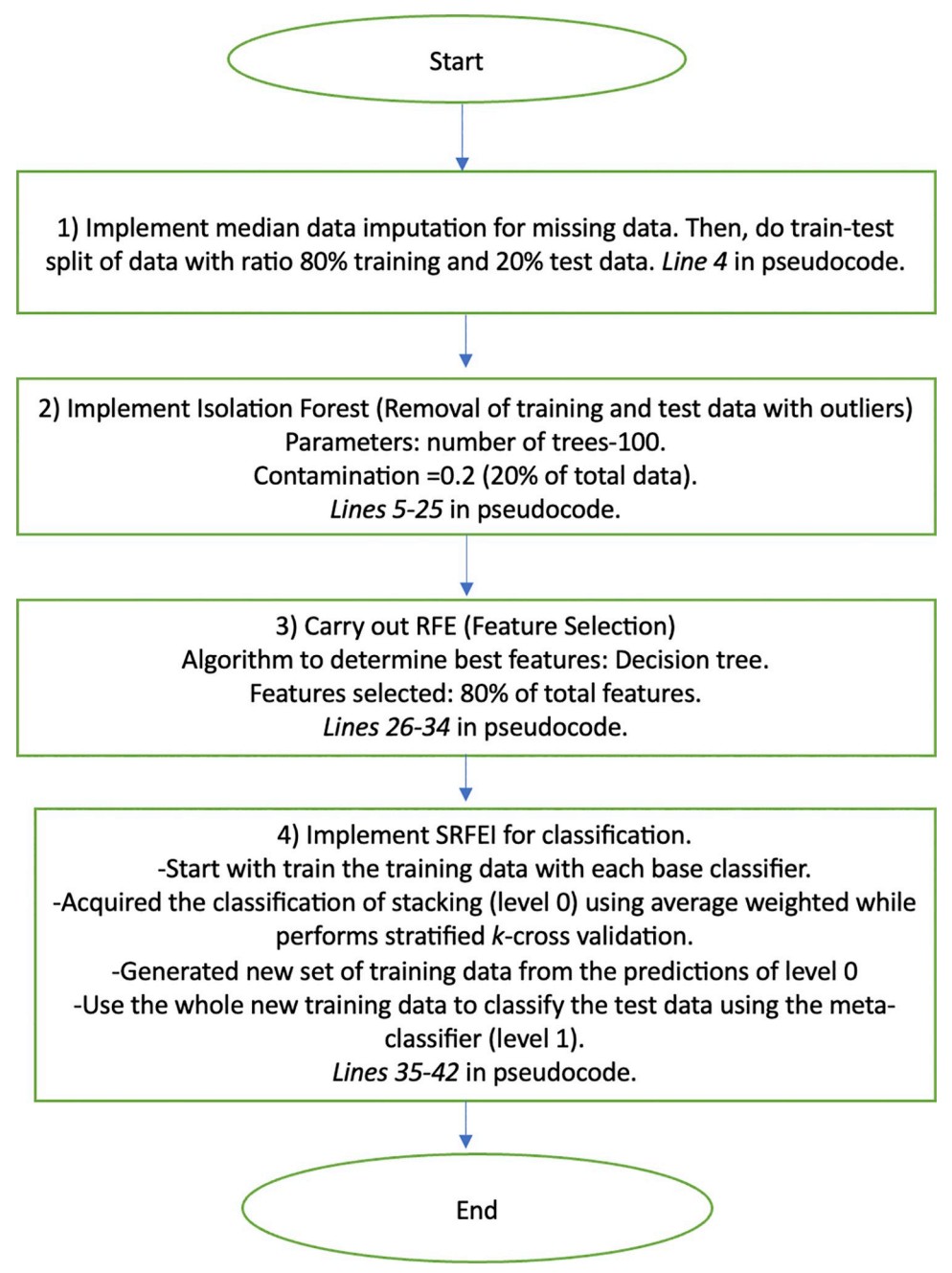

**Fig 2. The flowchart of SRFEI.**

resulting in an increment in accuracy from 76.363% (conventional stacking) to 79.077%, and the precision increased from 74.144 in stacking to 76.065. Similar improvements were observed in the Diabetes Prediction dataset. The training time was reduced from 8949.979 seconds to 4066.058 seconds, and the accuracy increased from 96.689% to 97.446%. The precision score increases from 93.952% to 95.199% when using SRFEI. Although the results slightly decreased in recall and F1-measure, accuracy and precision improved in both datasets. The higher accuracy score compared to stacking demonstrates the SRFEI's capability to predict all classes accurately. Then, the high precision value signifies the SRFEI capability to correctly

Pseudocode of SRFEI

| | |
|---|---|
| 1 | Inputs-training data $L$, Test data $J$, total number of trees $e$, subsampling ratio $d$, total feature |
| 2 | $X$, base classifiers $h$; |
| 3 | Output: classification of SRFEI algorithm $M$. |
| 4 | Step 1: Data imputation through median imputation and split of training ($L$) and test data ($J$); |
| 5 | Step 2: Start Isolation Forest $W$; |
| 6 | **For** $i$=1, to $e$ |
| 7 | Randomly subsample training data with ratio $d$; |
| 8 | Randomly select feature x from the subsampled data; |
| 9 | Select a random threshold $t$ from the range ($\min_{xi}$, $\max_{xi}$) for feature $x$; |
| 10 | Split the data into left and right branches based on selected features; |
| 11 | **If** $L_i < t$; |
| 12 | Link the obtained tree to the left tree; |
| 13 | **Else if** $L_i > t$; |
| 14 | Link the obtained tree to the right tree; |
| 15 | Continue growing the tree recursively until a predefined stopping condition; |
| 16 | Aggregate the anomaly score of the trees; |
| 17 | **For** $s$=1, to $S$ |
| 18 | Calculate the anomaly score for $L$; |
| 19 | Remove $L_s$ with the highest anomaly score; |
| 20 | **End;** |
| 21 | **For** $s$=1, to $S$ |
| 22 | Calculate the anomaly score for $J$; |
| 23 | Remove $J_s$ with the highest anomaly score; |
| 24 | **End** |
| 25 | **End;** |
| 26 | Step 3: Implement RFE $R$; |
| 27 | **For each** $X_i$ **in** $X$: |
| 28 | Train L with CART using the best features (eliminates one feature at a time); |
| 29 | Calculate performance; |
| 30 | Calculate feature importance (*Gini impurity*); |
| 31 | Keep the best features; |
| 32 | **End;** |
| 33 | Sort features based on cumulative importance or performance; |
| 34 | Select the top 80% features as the best feature U; |
| 35 | Step 4: Implement $M$ |
| 36 | **For** $y$=1 do $h$ |
| 37 | Learn $L_Y$ based on stacking using base classifiers (level 0) with $U$; |
| 38 | **End** |
| 39 | Combine predictions from base classifiers; |
| 40 | Create a new set of training data $N$ using the predictions of the ensemble; |
| 41 | Learn $M$ with $N$ as the meta-classifier stage (level 1) while predicting using $J$; |
| 42 | Return $M$; |

**Fig 3. The pseudocode of SRFEI.**

**Table 1. PIMA Indians diabetes dataset classification result.**

| Metrics | Results: Stacking | Results: SRFEI |
|---|---|---|
| Test accuracy (%) | 76.363 | 79.077 |
| Precision (%) | 74.144 | 76.065 |
| Recall (%) | 71.584 | 70.465 |
| F1 measure (%) | 72.297 | 71.865 |
| Training time (s) | 4.286 | 2.646 |
| SD | 3.010 | 2.139 |

**Table 2. Diabetes prediction dataset classification result.**

| Metrics | Results: Stacking | Results: SRFEI |
|---|---|---|
| Test accuracy (%) | 96.689 | 97.446 |
| Precision (%) | 93.952 | 95.199 |
| Recall (%) | 82.709 | 76.064 |
| F1 measure (%) | 87.304 | 82.535 |
| Training time (s) | 8949.979 | 4066.058 |
| SD | 0.206 | 0.189 |

identify individuals at risk of diabetes among those predicted to be positive. In the context of the diabetes domain, precision is particularly crucial due to the potential ramifications of mis-classifying individuals as positive when they do not have diabetes. Such mistakes can lead to many issues, such as unnecessary stress and unwarranted medical interventions. SRFEI managed to minimize the occurrence of false positives, as misclassifying non-diabetes cases as positive is deemed more unfavorable despite correctly identifying both cases being extremely important in the diabetes domain. Fig 4 illustrates the comparison of the PIMA Indians Diabetes dataset performance, while the comparison of the Diabetes Prediction dataset performance is illustrated in Fig 5.

Next, the statistical *t*-tests were conducted to assess the significance of the differences in results between conventional stacking and SRFEI. The experiment employed a two-tailed test, which is more widely accepted and avoids directional bias. The significance level in the hypothesis testing, denoted as alpha ($\alpha$), is set as 0.05 because this value is commonly applied in many scientific research [51]. A significance level of 0.05 means there is a 5% chance of incorrectly rejecting the null hypothesis. If the calculated *p*-value exceeds the predetermined significance level ($\alpha = 0.05$), the null hypothesis (H0), asserting no significant difference between stacking ensemble and SRFEI, is accepted. Conversely, if the *p*-value is less than $\alpha$, the

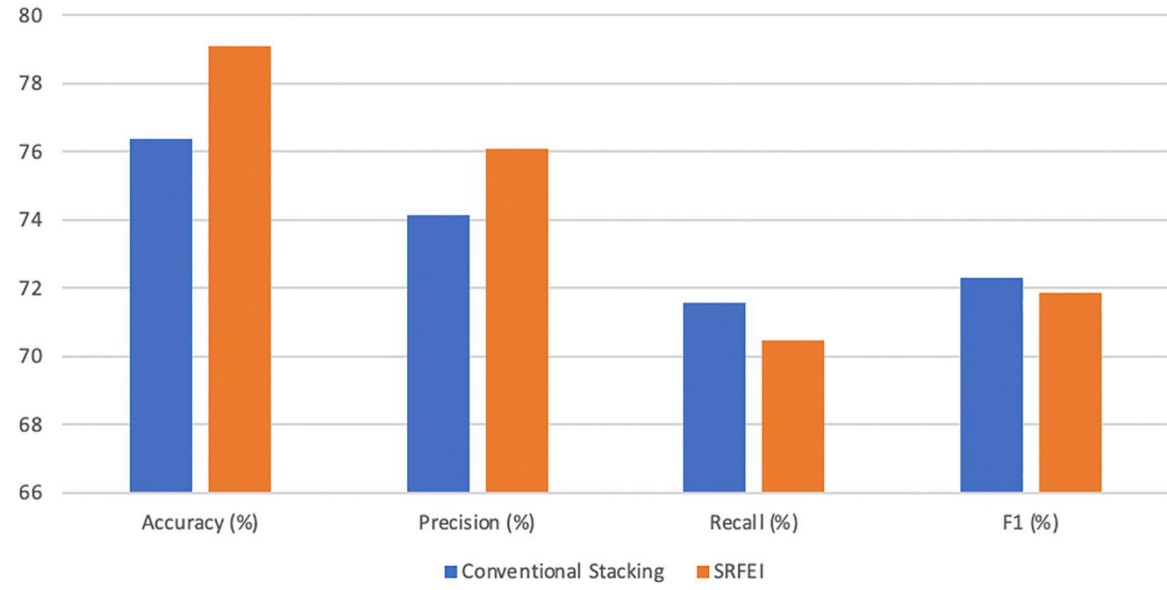

**Fig 4. PIMA Indians diabetes dataset performance comparison.**

Diabetes Prediction Dataset

**Fig 5. Diabetes prediction dataset performance comparison.**

alternative hypothesis (H1), indicating a significant difference, is accepted. The results used were thirty independent runs for both conventional stacking and SRFEI, which makes the degree of freedom 58 in both datasets. In the *t*-test of independent samples of both methods for the PIMA Indians Diabetes dataset classification, the *p*-value is 0.000824989, which is less than 0.05. Hence, this shows that the PIMA Diabetes dataset classification results difference is significant at $p < 0.05$. Meanwhile, in the t-test of independent samples of both methods for the Diabetes Prediction dataset classification, the *p*-value is 2.07399E-21, which is less than 0.00001. The results difference for the Diabetes Prediction dataset classification is also significant at $p < 0.05$. Thus, based on the *t*-test results, there are significant differences in accuracy increment between the conventional stacking and SRFEI.

## 4.2 Comparative analysis of the methods

The accurate prediction of diabetes is paramount for early detection and effective disease management. The objective of comparative analysis is to identify whether the SRFEI is comparable with other methods. Therefore, this study validates the method's performance by comparing it with existing state-of-the-art methods mainly through the classification result of the benchmark dataset, the PIMA Indians Diabetes. This study also evaluates the method's performance on the recently released Diabetes Prediction dataset. The analysis was done by employing the results of several eminent methods implemented using WEKA to establish a baseline for performance comparison. In alignment with other studies, accuracy was chosen as the benchmark evaluation metric. Tables 3 and 4 present the comparative analyses of the PIMA Indians Diabetes and Diabetes Prediction datasets classification results consecutively.

SRFEI acquired 79.077% of classification accuracy, surpassing the existing techniques such as Random Forest with the application of k-means clustering, PCA and importance ranking that obtained 75.22%, Gradient Boosting with 70%, Modified Bayes Network with 72.3%, XGBoost-SMOTE with 78.29%, and AdaBoost-SS with 73.88% using the PIMA Indians Diabetes dataset [17, 52–55]. For more validation, this study makes another comparison of the PIMA Indians Diabetes dataset classification that includes precision analysis. SRFEI outperforms Deep Neural Networks in terms of accuracy and precision, in which Deep Neural

**Table 3. Comparative analysis of the PIMA Indians diabetes dataset classification results.**

| Method | Accuracy (%) |
|---|---|
| Random forest- $k$-means clustering, principal component analysis (PCA), and importance ranking [17] | 75.22 |
| Gradient Boosting [52] | 70 |
| Modified Bayes Network [53] | 72.3 |
| XGBoost-SMOTE [54] | 78.29 |
| AdaBoost- Stability Selection (SS) [55] | 73.88 |
| **Proposed method (SRFEI)** | **79.077** |

Networks acquired an accuracy of 64.5% and precision of 64% using the PIMA Indians Diabetes dataset [56]. With the same dataset, SRFEI surpasses ANN-Sequential Forward Selection, which obtains an accuracy of only 78.41% and a precision score of 72.07% [57]. For the Diabetes Prediction dataset, this study applied three existing ensembles and one single classifier method for comparison to identify the method's capability. SRFEI, with 97.446% accuracy and 95.199% precision, surpassed the existing methods in Table 4, consisting of AdaBoost M1 and bagging with J48 as classifiers, Voting ensemble with Bayes net classifiers, LogitBoost with decision stump as classifiers, and Naïve Bayes. The voting ensemble acquired an accuracy of 96.672%, while LogitBoost acquired an accuracy of 97.187%. AdaBoost M1 acquired an accuracy of 96.275%, bagging obtained an accuracy of 97.033%, and lastly, Naïve Bayes acquired an accuracy of 94.113% with a precision of 94.2%. The analyses show that the SRFEI performs better than existing methods.

## 5. Discussion

In this study, the newly proposed method, SRFEI, was introduced and evaluated primarily for diabetes classification. The performance of SRFEI was compared against existing state-of-the-art methods using the PIMA Indians Diabetes and the recently released Diabetes Prediction dataset. The findings of the study demonstrated that SRFEI outperforms the existing methods on both datasets' classification results, displaying that SRFEI is a promising approach for diabetes prediction. At the same time, it can predict using a minimal number of features. The comparative analysis of the benchmark dataset, the PIMA Indians Diabetes dataset, and the newly released Diabetes Prediction dataset provide a standardized and comparable basis for methods evaluation. The choice of accuracy as the primary evaluation metric is common in classification tasks as it enables straightforward comparison with other existing studies.

The SRFEI method stands out due to the utilization of stacking, which integrates multiple base classifiers and a meta-classifier for the prediction. This integration of classifiers allows the model to benefit from the strengths of each classifier while alleviating the single classifiers' limitations. In level 0 of SRFEI, secondary data is generated from the prediction of base classifiers.

**Table 4. Comparative analysis of the diabetes prediction dataset classification results.**

| Method | Accuracy (%) |
|---|---|
| AdaBoost M1 (J48 classifier) | 96.275 |
| Bagging (J48 classifier) | 97.033 |
| Voting ensemble (Bayes net) | 96.672 |
| LogitBoost (Decision stump classifier) | 97.187 |
| Naïve Bayes | 94.113 |
| **Proposed method (SRFEI)** | **97.446** |

This secondary data becomes input features for the meta-classifier in level 1, which then makes the final prediction. The integration of multiple base classifiers assists in addressing the issue of overfitting. Overfitting generally transpires when a model becomes overly complex and excels on the training data but generalizes poorly on unseen data. As SRFEI combines the predictions from different base classifiers, it reduces the potential of overfitting by leveraging a diverse set of models as it learns different aspects of the data. SRFEI has superior performance compared to the existing methods, which shows that the SRFEI method has less of a tendency to overfit.

Another possible explanation for the competent performance of SRFEI can be attributed to the ability of the model to capture complex patterns in the data effectively. Three reasons cause this: firstly, due to aggregation of the predictions of base classifiers. Combining predictions from diverse classifiers and learning provides a more comprehensive view of the underlying data patterns, leading to more reliable predictions. Simultaneously, the utilization of the average weighted technique accentuates the cumulative distinctions among the base classifiers. Consequently, it ensures a diverse model composition by incorporating varying perspectives from each base classifier. Moreover, average weighted can help mitigate the impact of extreme predictions from individual models that can lead to a more stable and balanced prediction. Next, the utilization of Isolation Forest as an outlier detection is efficient for detecting the anomalies and outliers in data. Therefore, the outliers can be removed effectively, eliminating bias in the diabetes data. The RFE feature selection simplifies the representation of the data in the model by focusing on the most relevant features. The unnecessary, irrelevant, and redundant features were removed, making the classification process more proficient. Thus, the utilization of RFE and Isolation Forest leads to a more concise representation of the underlying data patterns. SRFEI managed to achieve the goal of this study, which is to reduce complexity and training time. The proposed method is highly efficient and can be accessed through reduced training time and improved accuracy and stability. The improved stability, which can be seen through the reduction of accuracy's SD values compared to conventional stacking, is due to the reduction of variations caused by irrelevant features. On top of everything, the combination of stacking, RFE, and Isolation Forest in the SRFEI method is compatible specifically with the diabetes datasets, which could have contributed to the method's relatively good classification performance on unseen data.

## 6. Conclusion

In conclusion, the study introduces the SRFEI method as a promising approach to diabetes classification to develop a method with high accuracy while using fewer features. The application of stacking, RFE, and Isolation Forest to form SRFEI ensures the elimination of irrelevant data in terms of data points and features. This, in turn, simplifies the ensemble method and expedites the training process, ultimately leading to reduced complexity. Through experimentation, the study identified that SRFEI exhibits comparable performance to other state-of-the-art methods on diabetes datasets in both accuracy and training time performance. While the study presents compelling evidence favoring SRFEI, several aspects must be considered for a comprehensive evaluation. Firstly, it would be beneficial to explore the interpretability of SRFEI, especially in the context of medical applications like diabetes prediction, where model interpretability is crucial for gaining the trust of healthcare professionals and patients. The assistance from the expert is helpful in this aspect. Moreover, to ensure the robustness and generalizability of the results, the SRFEI method should be tested on a more extensive range of diverse datasets. Another limitation of SRFEI is that it may not acquire the optimal hyperparameters during classification, which would be detrimental to the performance results in some

situations. Utilizing hyperparameter tuning techniques such as Genetic Algorithm and Grid Search would be useful in boosting the results, as the model can obtain optimal hyperparameters. Looking ahead, future research endeavors could also focus on more suitable data preprocessing and carefully selecting diverse base classifiers to enhance the results even further. The SRFEI method demonstrated outstanding potential and consistently performed well across various assessment metrics, including test accuracy, precision, recall, F1 measure, training time, and SD. Overall, the findings of this study showcase the effectiveness of SRFEI and its capability to handle datasets with reduced features, making it a valuable contribution to the field of machine learning. This study envisions SRFEI becoming a high-performing solution for various practical applications in diabetes.

## Author Contributions

**Conceptualization:** Nur Farahaina Idris.

**Data curation:** Nur Farahaina Idris.

**Formal analysis:** Nur Farahaina Idris, Mohd Arfian Ismail.

**Funding acquisition:** Mohd Arfian Ismail, Ashraf Osman Ibrahim, Anas W. Abulfaraj, Faisal Binzagr.

**Investigation:** Nur Farahaina Idris.

**Methodology:** Nur Farahaina Idris.

**Supervision:** Mohd Arfian Ismail, Mohd Izham Mohd Jaya.

**Validation:** Mohd Arfian Ismail.

**Visualization:** Nur Farahaina Idris.

**Writing – original draft:** Nur Farahaina Idris.

**Writing – review & editing:** Nur Farahaina Idris.

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
