## [Decision Letter · Decision Letter 0]

5 Mar 2024

PONE-D-23-44193Stacking with Recursive Feature Elimination -Isolation Forest for classification of Diabetes MellitusPLOS ONE

Dear Dr. Ismail,

Thank you for submitting your manuscript to PLOS ONE. After careful consideration, we feel that it has merit but does not fully meet PLOS ONE’s publication criteria as it currently stands. Therefore, we invite you to submit a revised version of the manuscript that addresses the points raised during the review process.

We look forward to receiving your revised manuscript.

Kind regards,

V. Vinoth Kumar

Academic Editor

PLOS ONE

 [The study was supported by the Fundamental Research Grant (RDU) with vot. No. RDU220304 from Universiti Malaysia Pahang Al-Sultan Abdullah].  

[The authors would also like to express gratitude to anyone who provided assistance, encouragement, or support during this research. The study was supported by the Fundamental Research Grant (RDU) with vot. No. RDU220304 from Universiti Malaysia Pahang Al-Sultan Abdullah. ]

 [The study was supported by the Fundamental Research Grant (RDU) with vot. No. RDU220304 from Universiti Malaysia Pahang Al-Sultan Abdullah]. 

Reviewers' comments:

Reviewer's Responses to Questions

**Comments to the Author**

1. Is the manuscript technically sound, and do the data support the conclusions?

Reviewer #1: Partly

Reviewer #2: Yes

Reviewer #3: Partly

2. Has the statistical analysis been performed appropriately and rigorously? 

Reviewer #1: No

Reviewer #2: No

Reviewer #3: Yes

3. Have the authors made all data underlying the findings in their manuscript fully available?

Reviewer #1: Yes

Reviewer #2: Yes

Reviewer #3: Yes

4. Is the manuscript presented in an intelligible fashion and written in standard English?

Reviewer #1: Yes

Reviewer #2: Yes

Reviewer #3: Yes

5. Review Comments to the Author

Reviewer #1: In this manuscript, the authors present a method called the Stacking Recursive Feature Elimination-Isolation Forest for diabetes prediction. However, the manuscript seems to have many limitations including the followings:

(i) There are many works in the domain. What is the specialty of your work over others? What is the motivation the work? The authors should clearly mention that.

(ii) It is very difficult to concentrate on the contributions. The authors should present the contributions in bullet form.

(iii) The authors should present a paragraph for presenting the organization of the work.

(iv) The authors should present the literature review in a separate section. Moreover, the authors should consider the recent works as well for that.

(v) What is the specialty of the materials and method section? Your presentation is very big but very general talks.

(vi) The authors should compare the works with other people's work.

(vii) Where is section 3?

Reviewer #2: The manuscript provides a comprehensive overview of the significance of diabetes mellitus as a global health concern and the integration of artificial intelligence, specifically stacking, to improve efficiency in diagnosis and healthcare delivery. The introduction effectively highlights the urgency of managing diabetes, emphasizing its impact on both the aging population and children. The proposal of a novel approach, Stacking Recursive Feature Elimination-Isolation Forest, to address the complexity in stacking is commendable.

However, the review suggests a need for more detailed explanations on the methodology, particularly regarding the integration of Recursive Feature Elimination and Isolation Forest.

Providing insights into the rationale behind this combination and its advantages over other methods would enhance the manuscript's clarity.

Additionally, the evaluation metrics utilized, such as accuracy, precision, recall, and F1 measure, are appropriate for assessing model performance.

However, a more thorough discussion on the limitations and potential biases of the proposed model is encouraged. Moreover, the manuscript could benefit from a comparative analysis with existing models in the field to establish the uniqueness and effectiveness of the proposed approach.

Overall, with improvements in methodology clarity and additional insights into limitations, this manuscript holds promise in contributing to the field of diabetes prediction using artificial intelligence.

Reviewer #3: The authors propose a Stacking Recursive Feature Elimination-Isolation Forest, to enhance diabetes prediction by reducing complexity and training time. This method utilizes Recursive Feature Elimination for efficient model design and Isolation Forest for outlier removal. Performance evaluation based on accuracy, precision, recall, F1 measure, and training time shows promising results, achieving 79.077% and 97.446% accuracy for PIMA Indians Diabetes and Diabetes Prediction datasets, respectively. The model has so many potentials, however, i have the following concrns/suggesions:

- The introduction may highlight more recent ML/AI similar model in the application of diabetes. I suggest to highlight PMID: 38132885 and more similar models.

- The methods are poorly written. The sections may include equations and more figures.

- The authors mentioned many times the importance of avoiding overfitting and underfitting without trying to explain how to avoid it (other than 1-fold CV). I suggest to address that by looking at the validation performance (testing data0 versus the training perfomance.

-The aucro for this kind of models is required accross different running points to show the overall performance.

6. PLOS authors have the option to publish the peer review history of their article (what does this mean?). If published, this will include your full peer review and any attached files.

Reviewer #1: No

Reviewer #2: **Yes: **SANGEETHA S K B

Reviewer #3: **Yes: **Abedalrhman Alkhateeb

---

## [Author Response · Author response to Decision Letter 0]

23 Mar 2024

Reviewer #1: In this manuscript, the authors present a method called the Stacking Recursive Feature Elimination-Isolation Forest for diabetes prediction. However, the manuscript seems to have many limitations including the followings:

(i) There are many works in the domain. What is the specialty of your work over others? What is the motivation the work? The authors should clearly mention that.

Reply: Thank you for the review. We have incorporated the contributions in the introduction section to elucidate the uniqueness of our work. Our method stands out due to its lower complexity compared to other ensemble techniques while maintaining effectiveness even with a reduced number of features (lines 405-408 and 682-684). Notably, it demonstrates high accuracy, specifically within the domain of diabetes (lines 685-687). Then, the motivation behind our work stems from the observation, which had been mentioned in the abstract and introduction, that while stacking is commonly employed in diabetes research, there is limited discussion regarding improvements in terms of complexity reduction (lines 25 and 81-82). By addressing this aspect, our research fills an important gap in the literature. 

(ii) It is very difficult to concentrate on the contributions. The authors should present the contributions in bullet form.

Reply: Contributions in bullet form have been added in the introduction section for clarity in lines 88-92 in track changes.pdf.

(iii) The authors should present a paragraph for presenting the organization of the work.

Reply: Thank you; to address your concern, it is in the introduction section for the organization of work. It is stated briefly to avoid repetition and redundancy in the body of knowledge later.

(iv) The authors should present the literature review in a separate section. Moreover, the authors should consider the recent works as well for that.

Reply: The literature review is in section 2 (Review of literature). Each subsection would explain the details of the elements of this study. This way, it is easier to see the related topics, like classification, ensemble, stacking (subsection of the ensemble), RFE, and isolation forest.

(v) What is the specialty of the materials and method section? Your presentation is very big but very general talks.

Reply: Noted. The pseudocode and flowchart were added to explain the proposed methodology. Additional step-by-step breakdowns were added to clarify how the algorithm works. Aside from that, more explanation had been added throughout the method section. You may see the pseudocode figure caption is on line 471, while the flowchart caption is on line 470.

(vi) The authors should compare the works with other people's work.

Reply: Noted; we have addressed your concern. The comparison had been made through comparative analyses in the tables and explanations, especially for the PIMA Indians diabetes dataset (benchmark data), which you can see on line 598. For the Diabetes prediction dataset, which is relatively new data so, we compared with AdaBoost M1 and bagging with J48 as classifiers, Voting ensemble with Bayes net classifiers, LogitBoost with decision stump as classifiers, and Naïve Bayes in which this method we conducted the experiments ourselves. This is explained in the comparative analysis of the methods section, starting on line 601.

(vii) Where is section 3?

Reply: Section 3 is the method section, which explains the methodology of the method. You can see this started in line 362 in track changes.pdf.

Reviewer #2: The manuscript provides a comprehensive overview of the significance of diabetes mellitus as a global health concern and the integration of artificial intelligence, specifically stacking, to improve efficiency in diagnosis and healthcare delivery. The introduction effectively highlights the urgency of managing diabetes, emphasizing its impact on both the aging population and children. The proposal of a novel approach, Stacking Recursive Feature Elimination-Isolation Forest, to address the complexity in stacking is commendable.

However, the review suggests a need for more detailed explanations on the methodology, particularly regarding the integration of Recursive Feature Elimination and Isolation Forest.

Providing insights into the rationale behind this combination and its advantages over other methods would enhance the manuscript's clarity.

Reply: Thank your for the suggestion. Modifications have been made to provide insights into the rationale behind this combination and its advantages over other methods. The contributions of this paper have been defined in the abstract, introduction, method, discussion, and conclusion properly. Some modifications have been made to clarify the research's contributions, mainly in the sections of Recursive Feature Elimination and Isolation Forest.

Additionally, the evaluation metrics utilized, such as accuracy, precision, recall, and F1 measure, are appropriate for assessing model performance. However, a more thorough discussion on the limitations and potential biases of the proposed model is encouraged. Moreover, the manuscript could benefit from a comparative analysis with existing models in the field to establish the uniqueness and effectiveness of the proposed approach.

Reply: Thank you for the suggestion. The limitations and potential biases of the model have been added in the conclusion section for further research work from lines 713 until 722. Lines 717-721 stated that the method may not obtain the optimal hyperparameters, which can affect the results. The study has added additional comparison in the comparative analyses section to strengthen the proof that our study managed to obtain better results than existing works on lines 598-603,

Overall, with improvements in methodology clarity and additional insights into limitations, this manuscript holds promise in contributing to diabetes prediction using artificial intelligence.

Reply: Noted, and we followed the advice. Firstly, we added additional step-by-step breakdowns to increase the clarity of how the algorithm works in the pseudocode figure. You may see the pseudocode figure caption is on line 471, while the flowchart caption is on line 470. Furthermore, the conclusion section has been improved and modified, as the limitations of the developed method have also been discussed in the conclusion section.

Reviewer #3: The authors propose a Stacking Recursive Feature Elimination-Isolation Forest, to enhance diabetes prediction by reducing complexity and training time. This method utilizes Recursive Feature Elimination for efficient model design and Isolation Forest for outlier removal. Performance evaluation based on accuracy, precision, recall, F1 measure, and training time shows promising results, achieving 79.077% and 97.446% accuracy for PIMA Indians Diabetes and Diabetes Prediction datasets, respectively. The model has so many potentials, however, i have the following concrns/suggesions:

- The introduction may highlight more recent ML/AI similar model in the application of diabetes. I suggest to highlight PMID: 38132885 and more similar models.

Reply: Thank your for the information. Modifications have been made to highlight different models. Other studies have been added to the manuscript, including the listed work recommended by the reviewers in which the citation is in line 66.

- The methods are poorly written. The sections may include equations and more figures.

Reply: Noted. The pseudocode and flowchart figures were added to explain the proposed methodology. Additional step-by-step breakdowns were added to increase the clarity on how the algorithm works. You may see the pseudocode figure caption is on line 471, while the flowchart caption is on line 470.

- The authors mentioned many times the importance of avoiding overfitting and underfitting without trying to explain how to avoid it (other than 1-fold CV). I suggest to address that by looking at the validation performance (testing data0 versus the training perfomance.

Reply: Avoiding overfitting is one of the advantages of the stacking ensemble. The term is to highlight the importance of stacking utilization compared to other machine learning methods. Furthermore, the capability to overcome overfitting can be seen as it acquires relatively high results in test data classification. The actual testing vs training accuracy results range from +/-4 for both datasets in each run, which is relatively good. However, the study wants to highlight the capability of the method to acquire better results with fewer features and its efficiency in time more than overfitting/underfitting issues. Thus, we decided not to address this in the paper.

-The aucro for this kind of models is required accross different running points to show the overall performance.

Reply: Thank you for your reminder. This technique has been used in this study as this model has been run across many points to acquire the overall performance. It had been run under thirty independent runs (as stated in the manuscript in lines 481 and 554 in track changes.pdf) in which it go with different seeds each time. Using different seeds, which are random number generators, influences various aspects of the training process, such as the shuffling of datasets or the selection of random samples during training.

---

## [Decision Letter · Decision Letter 1]

9 Apr 2024

Stacking with Recursive Feature Elimination -Isolation Forest for classification of Diabetes Mellitus

PONE-D-23-44193R1

Dear Dr. Ismail,

We’re pleased to inform you that your manuscript has been judged scientifically suitable for publication and will be formally accepted for publication once it meets all outstanding technical requirements.

Kind regards,

V. Vinoth Kumar

Academic Editor

PLOS ONE

Additional Editor Comments (optional):

Reviewers' comments:

Reviewer's Responses to Questions

**Comments to the Author**

1. If the authors have adequately addressed your comments raised in a previous round of review and you feel that this manuscript is now acceptable for publication, you may indicate that here to bypass the “Comments to the Author” section, enter your conflict of interest statement in the “Confidential to Editor” section, and submit your "Accept" recommendation.

Reviewer #2: All comments have been addressed

Reviewer #3: All comments have been addressed

2. Is the manuscript technically sound, and do the data support the conclusions?

Reviewer #2: Yes

Reviewer #3: Yes

3. Has the statistical analysis been performed appropriately and rigorously? 

Reviewer #2: Yes

Reviewer #3: Yes

4. Have the authors made all data underlying the findings in their manuscript fully available?

Reviewer #2: Yes

Reviewer #3: Yes

5. Is the manuscript presented in an intelligible fashion and written in standard English?

Reviewer #2: Yes

Reviewer #3: Yes

6. Review Comments to the Author

Reviewer #2: (No Response)

Reviewer #3: The authors have adequatly addressed the reviewer's comments. I endorse publishing this manuscript.

7. PLOS authors have the option to publish the peer review history of their article (what does this mean?). If published, this will include your full peer review and any attached files.

Reviewer #2: **Yes: **SANGEETHA S K B

Reviewer #3: **Yes: **Abedalrhman Alkhateeb
